# Synthesis, Pharmacological and Structural Characterization of Novel Conopressins from *Conus miliaris*

**DOI:** 10.3390/md18030150

**Published:** 2020-03-06

**Authors:** Julien Giribaldi, Lotten Ragnarsson, Tom Pujante, Christine Enjalbal, David Wilson, Norelle L. Daly, Richard J. Lewis, Sebastien Dutertre

**Affiliations:** 1Institut des Biomolécules Max Mousseron, UMR 5247, Université de Montpellier-CNRS, 34095 Montpellier, France; julien.giribaldi@umontpellier.fr (J.G.); tom.pujante@etu.umontpellier.fr (T.P.); christine.enjalbal@umontpellier.fr (C.E.); 2Institute for Molecular Bioscience, The University of Queensland, St Lucia, Queensland 4072, Australia; l.ragnarsson@imb.uq.edu.au (L.R.); r.lewis@uq.edu.au (R.J.L.); 3Centre for Molecular Therapeutics, Australian Institute of Tropical Health and Medicine, James Cook University, Cairns, QLD 4878, Australia; david.wilson4@jcu.edu.au (D.W.); norelle.daly@jcu.edu.au (N.L.D.)

**Keywords:** conopressin, vasopressin, venom, cone snail, conotoxin

## Abstract

Cone snails produce a fast-acting and often paralyzing venom, largely dominated by disulfide-rich conotoxins targeting ion channels. Although disulfide-poor conopeptides are usually minor components of cone snail venoms, their ability to target key membrane receptors such as GPCRs make them highly valuable as drug lead compounds. From the venom gland transcriptome of *Conus miliaris*, we report here on the discovery and characterization of two conopressins, which are nonapeptide ligands of the vasopressin/oxytocin receptor family. These novel sequence variants show unusual features, including a charge inversion at the critical position 8, with an aspartate instead of a highly conserved lysine or arginine residue. Both the amidated and acid C-terminal analogues were synthesized, followed by pharmacological characterization on human and zebrafish receptors and structural investigation by NMR. Whereas conopressin-M1 showed weak and only partial agonist activity at hV1bR (amidated form only) and ZFV1a1R (both amidated and acid form), both conopressin-M2 analogues acted as full agonists at the ZFV2 receptor with low micromolar affinity. Together with the NMR structures of amidated conopressins-M1, -M2 and -G, this study provides novel structure-activity relationship information that may help in the design of more selective ligands.

## 1. Introduction

Cone snail venoms represent a unique source of hundreds of thousands of bioactive peptides, yet it is estimated that less than 1% of this diversity has been pharmacologically characterized. Nevertheless, the therapeutic potential of *Conus* venom is undeniable, with one peptide approved by the FDA for the treatment of chronic pain [1,2,3], and several others that are in clinical trials. Furthermore, these natural peptides have become indispensable tools to decipher the physiological role and function of various ion channel subtypes and other membrane receptors [4,5]. *Conus* peptides are divided into two mains groups, based on the number of disulfide bonds; the disulfide-poor conopeptides (0 or 1 disulfide bond) and the disulfide-rich conotoxins (two or more disulfide bonds) [6]. Disulfide-poor conopeptides are usually minor components of cone snail venom, but interestingly several subclasses, such as the contulakins or ρ-conopeptides, target G protein-coupled receptors (GPCRs) [6]. Although less abundant than the conotoxins that target ion channels, recent venomics analyses of venom glands have revealed these disulfide-poor conopeptides remain an untapped resource of novel bioactives.

The first vasopressin-oxytocin-related conopeptides, Lys-conopressin-G and Arg-conopressin-S, were isolated more than thirty years ago from the venom of two piscivorous species, *Conus geographus* and *Conus striatus*, respectively [7]. Intracerebral injection into mice induced similar effects compared to neurohypophysal hormone injection, and therefore, it was hypothesized that conopressin-G/S might act as agonists of the same class of GPCR in the brain. While only one subtype of OTR (oxytocin receptor) has been identified, three different vasopressin receptor subtypes have been characterized that possess different pharmacological and G protein-coupling properties [8]. V2 receptors (V2Rs) are coupled to adenylyl cyclase and their activation produces an antidiuretic effect via the kidneys. V1aR and V1bR, as well as OTR are coupled to phospholipase C but produce different biological effects upon activation: the V1aR regulates blood pressure and vasoconstriction, whereas the V1bR is responsible for the corticotrophin release from the pituitary gland. Compounds targeting the OTR are used clinically to stimulate the contraction of uterine and mammary myocytes [9,10]. Interestingly, unlike vasopressin and oxytocin, conopressin-G and -S display an additional positive charge at position 4, which is only found in two other endogenous vasopressin analogues, namely cephalotocin (*Octopus vulgaris*) and annetocin (*Eisenia foetida*) [7,11]. Lys-conopressin-G was later isolated from other non-venomous snail species and was therefore proposed as an endogenous invertebrate vasopressin analog. Consequently, an underlying question is to determine whether conopressins are simply endogenous or true venom peptides in cone snails, since a role in prey capture has never been demonstrated. However, given that more conopressin variants have been identified in different venomous cone snail species, it clearly suggests they are venom peptides with the typical diversification observed in conotoxins [12]. Among them is the unique γ-conopressin-vil purified from *Conus villepinii* venom, which displays a carboxyglutamate residue conferring the capacity of the peptide to change its conformation in the presence of calcium ions [13]. Another unique example is conopressin-T isolated from *Conus tulipa*, which, unlike vasopressin-oxytocin related peptides, is a selective antagonist of V1a receptors [11]. Interestingly, position 9 was described as an antagonist switch and replacement of glycine at this position by a valine residue turns oxytocin and Arg-Vasopressin (AVP) from full agonists to full antagonists. The L7P-conopressin-T mutant displays an increased affinity for the V1a receptors, whereas activity at the V1b and V2 receptors remains unchanged, suggesting a favorable V1aR selective conformational change induced by the proline residue [11]. 

Docking studies on a three-dimensional model of the V1a receptor revealed that Arg-vasopressin binds into a 15‒20 Å deep cleft defined by the transmembrane helices of the receptor. Residues located in this region that interact with agonist ligands are highly conserved in all the vasopressin and oxytocin receptors suggesting the agonist-binding pocket is common to all the different subtypes of this receptor family [11]. Interestingly, mutations of residues located in the receptor agonist binding site do not affect antagonist activity, suggesting a different binding mode for antagonist ligands [11]. Indeed, the different binding sites of various antagonist ligands are formed by transmembrane helices 1, 2 and 7, whereas the agonist binding site is mainly made by the three extracellular domains of the oxytocin receptor, as evidenced by Postina et al. [14]. Based on the bovine rhodopsin structure, three-dimensional molecular models of the V1aR and V1bR complexed with vasopressin suggested that four key residues fine tune the binding of vasopressin and related peptide agonists to both receptor subtypes [15]. While these predictions have been validated by receptor mutants and may enable the design of V1a and V1b receptor selective agonists, more information on structure-activity relationships of novel ligands are highly desirable [15]. 

In this study, we report on the synthesis, pharmacological characterization and structure of two new conopressin-related peptides identified in *Conus miliaris* transcriptome. All peptides were characterized on both human and zebrafish receptors using conopressin-G, vasotocin, oxytocin and vasopressin as controls. Together with their NMR structures and considering their unique sequences, this study provides valuable structure-function information that might be useful to guide the design of new vasopressin receptor selective ligands. 

## 2. Results

Two new conopressin-related sequences were identified in the venom gland transcriptome of the vermivorous species *Conus miliaris*. Both sequences had the characteristic mature nonapeptide containing one disulfide bond forming a six-membered ring plus three exocyclic residues. Most surprisingly, both sequences displayed very unusual substitutions, including an aspartic acid instead of the highly conserved basic residue at position 8, a serine instead of the highly conserved amidated glycine at the C-terminal (C-ter), and conopressin-M1 has also a proline residue in position 3 (Figure 1A). Considering the positions of these unusual substitutions, including in the functionally critical exocyclic region, it was of interest to investigate the pharmacological and structural properties of these novel conopressins.

### 2.1. Chemical Synthesis 

All conopressins discovered to date have an amidated C-terminal, yet the precursors of the conopressin-M1/M2 sequences do not display the usual G_10_K_11_R_12_ motif (data not shown), which is a typical enzymatic recognition site where the glycine residue at position 10 is enzymatically converted to a C-terminal amide group [20]. Therefore, without MS evidence for one or the other, we synthesized both C-terminal amide (Con-M1/M2am) and acid (Con-M1/M2ac) versions of the peptides. Con-G from the piscivorous *C. geographus* was also synthesized as its three-dimensional structure is unknown and its pharmacological characterization mostly incomplete. After RP-HPLC purification of the folded peptide, the homogeneity was assessed by analytical RP-HPLC and MS (Figure 1B). Pure peptide yields were nearly two-fold better for the amidated peptides (around 30% compared to acid peptides around 17%), which can be attributed to the better stability of the amide versus the ester bond to the linker. The Con-G yield was up to 70%, however, and it is difficult to conclude if this difference arises from a better folding yield or a better synthesis yield, since linear products have not been isolated and purified prior to oxidation given the size of the peptides. 

### 2.2. NMR Spectroscopy 

One-dimensional and two-dimensional NMR spectra were recorded on the conopressin peptides to provide insight into the solution structures. The spectra of Con-M1 contain additional peaks, indicative of multiple conformations. By contrast, Con-G and Con-M2 display predominately one set of peaks, indicating that divergent conformations are not present in solution for these peptides. Despite the presence of multiple conformations for the Con-M1 peptides, the major conformation could be assigned. Comparison of the chemical shifts between the amide and acid forms of Con-M1 and Con-M2 shows that the state of the C-terminal residue does not impact the structure as the chemical shifts are very similar between the two peptides as shown in Figure 2. 

Given the similarity between the acid and amide forms of the peptides, the three-dimensional structures of only the amide forms of Con-M1 and Con-M2 were calculated. The three-dimensional structure of Con-G (containing a C-terminal amide) was also determined. Despite the small size of these peptides, relatively well-defined structures could be determined, with calculated RMSD for the 20 best structures (values for backbone residues) of 0.895, 0.676 and 1.664 Å for Con-G, Con-M1 and Con-M2 respectively. The lower RMSD value of Con-M1 could be attributed to the proline residue in position 3 which seems to further constrain the backbone (Figure 2). Overall, the structures appear slightly different to the previously described Con-T [11], where the three exocyclic residues are more flexible (Figure 2) due to the absence of the conserved and structurally constrained proline residue in position 7. The structures of the three peptides are similar and they do not contain regular secondary structure but rather a turn region formed by the disulfide bond between Cys1 and Cys6. Deuterium exchange experiments can provide insight into residues likely to be involved in hydrogen bonds. 1D and TOCSY spectra were recorded following dissolution of the peptides in 100% D_2_O and based on these experiments Con-M1 and Con-M2 display slowly exchanging amide protons (at least 4-6 residues). The majority of peaks are exchanged within 30 minutes but the detection of amide protons in the D_2_O solution, particularly for such small peptides, suggests that the amide protons are protected from the solvent to some extent and are likely to be involved in hydrogen bonds that stabilize the structures. By contrast, the amide peaks of Con-G are exchanged within 10 minutes and could not be identified using TOCSY spectra. 

### 2.3. Pharmacological Characterization on Human and Zebrafish Receptors

We performed a fluorescent imaging plate reader (FLIPR) Ca^2+^ mobilization assay and a second-messenger cyclic adenosine monophosphate (cAMP) assay (for V2R) to determine the biological activity of the four new conopressins and Con-G. Considering this structure-activity relationship (SAR) study from a drug design perspective, we investigated the agonist and antagonist activity of the conopressins on the human oxytocin receptor (hOTR), human vasopressin-1a receptor (hV1aR), human vasopressin-1b receptor (hV1bR) and human vasopressin2 receptor (hV2R). However, since *Conus geographus* is a piscivorous cone snail feeding on fish, it was of interest to also investigate conopressin activities on zebrafish (*Danio rerio*) receptors, namely zebrafish vasopressin1-a1 receptor (ZF V1a1R), zebrafish vasopressin1-a2 receptor (ZF V1a2R), zebrafish vasopressin2 receptor (ZF V2R) and zebrafish oxytocin/isotocin receptor (ZF oxy/isoR) (Figure 3 and Figure 4). Oxytocin, vasopressin and vasotocin were used as reference compounds. 

Overall, we can delineate roughly three classes of potency: (i) reference compounds displaying EC_50_ values in the picomolar/nanomolar range, (ii) Con-G exhibiting potencies in the high nanomolar range and iii) *C. miliaris* conopressins that mostly show no or weak activity, except at the ZF V2R with an EC_50_ in the micromolar range. The micromolar range affinity displayed by the *C. miliaris* conopressin peptides clearly emphasize the negative influence of the conserved residue substitutions, which will be further discussed. Interestingly, we demonstrate for the first time that Con-G is more active on fish receptors than on their human counterparts, which supports an evolved role of Con-G in the envenomation process, although more direct evidence is required. Con-G acts as a partial agonist at hOTR, ZF V1a2R and ZF oxy/isoR with 28%, 69% and 62% response of control respectively, but as a full agonist at all other receptors. Con-M1am is the only *C. miliaris* peptide showing a weak and partial agonist response at hV1b and hV1a with 26% and 58% response of control, respectively. Somehow expectedly, endogenous mammalian peptides (oxytocin and vasopressin) are more active than the non-mammalian vasotocin peptide at all human receptors except for the OTR. Inversely, vasotocin (fish endogenous ligand) was more active than mammalian peptides at all zebrafish receptors, which is consistent from an evolutionary point of view. The EC_50_ values are reported in Table 1. There was no antagonist activity detected for any of the synthesized conopressins up to 10 µM and control compounds did not induce a response in untransfected cells, which confirms that our responses are receptor specific (data not shown).

## 3. Discussion

Conopressins are vasopressin-like peptides originally found in the venom of piscivorous cone snails (Con-G from *C. geographus* and Con-S from *C. striatus*) [7]. Lys-conopressin G was later found in the venom of the vermivorous *C. imperialis* as well as other non-venomous snails, suggesting that it may represent an endogenous peptide [21]. However, more recent reports describing divergent sequences of conopressins such as γ-conopressin-vil or Con-T support their role as venom peptides [11,13]. In this study, we describe two novel vasopressin-like sequences retrieved from the venom gland transcriptome of the vermivorous *Conus miliaris*, conopressin-M1 and -M2. Given the unusual sequence deviation of these conopressins compared to currently known oxytocin/vasopressin related peptides, (i.e., proline residue at position 3 of Con-M1, glycine at position 4, negatively charged aspartic acid at position 8 in place of conserved basic residue, and the lack of terminal glycine residue (Figure 1A)), it was of interest to investigate their structure and pharmacological profile. It has previously been demonstrated that a glycine substitution by a valine residue at position 9 will switch oxytocin and AVP from agonist to antagonist [11]. Using chemical synthesis, the C-terminal acid and amide of each of the novel conopressin peptides (Con-M1ac/am, Con-M2ac/am) were obtained with good purity and in sufficient yield to perform pharmacological and structural characterization. Since the pharmacological profile of conopressin-G is still poorly characterized and its structure unknown, this peptide was also synthesized for comparative purposes. Interestingly, Con-M1 did not resolve as a sharp UV peak but instead two partially separated peaks displaying the same mass. Possibly, the additional proline residue at position 3 of Con-M1 peptides may cause dynamic conformational exchanges between cis-trans isomerization, a conformational heterogeneity that has been described in other conotoxins [16,17,18]. This explanation is consistent with the NMR spectra of the Con-M1 peptides, which display the presence of multiple conformations. 

Pharmacological characterization of all peptides was carried out on human AVP and OT receptors and for the first time also on zebrafish receptors. Besides AVP and OT, vasotocin (fish endogenous ligand) was also tested as a reference. Not surprisingly, AVP and OT show higher affinity for human receptors compared to vasotocin, whereas the opposite was true on fish receptors. The only exception concerns a slightly more active vasotocin (1.62 nM) at human OTR compared to OT (4.57 nM) and AVP (8.86 nM). Contrasting results were obtained regarding the activity of the conopressins. Con-G, which possesses a basic residue in position 8, acts as a full agonist at all human receptors, except OTR. Con-G was as potent as oxytocin on hV1aR and hV1bR, with EC_50_ in the high nM range (52-123 nM), yet it displays lower affinity at the hV2R (300 nM). Higher affinities were achieved for Con-G at the zebrafish receptors, particularly at the ZF V1a1R (EC_50_ = 10 nM). As anticipated from the absence of a basic residue at position 8, *C. miliaris* conopressins showed no or weak activity at all human receptors. Only the amidated form of Con-M1 acted as partial agonist at the hV1AR and hV1bR. However, both the acid and amidated Con-M2 were full agonists at the ZF V2R, with affinities in the low μM range (1.7-3.6 μM). 

In terms of SAR information from this work, it seems evident that an aromatic residue at position 3 improves the peptide selectivity for hV1aR and hV1bR, mostly by decreasing the potency at all other receptors, especially ZF oxy/isoR. Furthermore, the presence of a basic residue instead of highly conserved glutamine residue at position 4 reduces the potency at all tested receptors, particularly at ZF oxy/isoR and hOTR, but even more drastically at hV2R. In order to gain insights into the interaction of the peptides ligands with the receptors, Rodrigo et al. [15] built three-dimensional molecular models of the complexes between AVP and the two receptor subtypes V1a and V1b based on the X-ray structure of bovine rhodopsin. Their predictions have been confirmed by directed mutagenesis studies and four key residues were identified that finely tune the binding of vasopressin and related peptide agonists to both receptor subtypes. Indeed, Glu^1.35^ and Asp^2.65^ residues are described as key anchoring residues to Arg^8^, which is evidenced by higher EC_50_ values of oxytocin, and similarly, explains the lack of activity of the *C. miliaris* conopressins that do not display a basic residue in position 8. 

Interestingly, vasopressin and vasotocin are nearly 10,000-fold more potent on hV2R than on ZF V2R (0.5 nM vs 5 μM), where they display EC_50_ values similar to oxytocin (16 μM), suggesting a crucial role of the basic residue in position 8 to enable a tight interaction with hV2R but not with ZF V2R. An alignment of hV2R and ZF V2R sequences reveals that all acidic residues important for ligand binding located in the loop between TMVI and TMVII of hV2R are not conserved in ZF V2R (Figure 5). Rodrigo et al. also identified Val^4.31^ and Pro^5.35^ as a hydrophobic subsite specific of V1bR according to its high affinity with d[Cha^4^]AVP [15,22]. The latter probably partly explains why vasopressin, which has a hydrophobic Phe residue at position 3, displays a slightly better affinity at hV1b compared to vasotocin (Table 1). A previous study from Mouillac et al. has shown that hydrophobic regions of the peptide hormones are accommodated by a hydrophobic pocket lying deep in the 7-TM (transmembrane) domain delineated between TMIII and TMVI of V1aR [9,15], confirming the slightly better affinity at hV1a compared to vasotocin. A tight hydrogen bond network also contributes to crucial interactions with V1aR and V1bR. Indeed, conserved Gln residues located at the rim of the cavity form hydrogen bond (H-bond) with side chains of Gln^4^ and Asn^5^ of the peptide hormones [15]. As a result, substitution of the vasotocin glutamine residue at position 4 by an arginine residue in Con-G leads to a loss of potency at hV1aR and hV1bR. Moreover, SAR studies show that a glycine carboxamide moiety is required for biological activity, enabling H-bonds between AVP/OT and Gln^214^, Gln^218^ of V1aR [9,23] and providing another plausible explanation for the lack of activity of *C. miliaris* conopressins, where the terminal glycine residue is replaced by a serine residue. 

In conclusion, the low activity on all tested receptors of Con-M1 and Con-M2 can be explained by (i) the substitution of Gln^4^ by a glycine residue, (ii) the absence of a basic residue in position 8, and (iii) the missing glycine residue in position 9. Overall, *C. miliaris* conopressins mostly show no or reduced activity on all receptor tested, except at the ZF V2R where EC_50_ values are in the micromolar range. Yeganeh et al. [24] showed that residues involved in the binding site are W293, W296, D297, A300, and P301. Our results suggest that there is an interaction between residue in position 8 of the ligand and D297 of the hV2R since the activity decreases (compared to vasotocin) for all ligands that do not display a positively charged residue in position 8. Interestingly, in the ZF V2R sequence, the equivalent of residue D297 is substituted by a serine residue (Figure 5), significantly reducing the potential electrostatic repulsion with the aspartic residue in position 8 of the *C. miliaris* conopressins, hence the detected activity of *C. miliaris* conopressins on ZF V2R over hV2R. From more evolutionary considerations, it is interesting to note that Con-G is more active on zebrafish (*Danio rerio*) receptors than on the human counterpart receptors, which is consistent with the piscivorous behavior of *C. geographus* and suggests a role for Con-G in the envenomation process. Given that *C. miliaris* is a worm hunting cone snail (preys on Polychaeta worms), the interspecies receptor molecular differences (worm/fish) may also account for the seemingly weak activities at zebrafish/human receptors of conopressin-M1 and -M2. Supporting this hypothesis, among the neurohormones identified in the genome of the Polychaeta worm *Capitella teleta* is an endogenous conopressin-like sequence devoid of basic residue at position 8 [25]. Future investigations should include a phylogenetic screening of conopressins based on the diet of the species it was isolated from to better reflect on the true biological activities of these venom peptides.

## 4. Materials and Methods 

### 4.1. Abbreviations

Acm, acetamidomethyl; ACN, acetonitrile; Boc, tert-butoxycarbonyle; DCM, Dichloromethane; DIPEA, diisopropylethylamine; DMF, N,N’-dimethylformamide; DTP, 2,2’-Dithiopyridine; ESI-MS, electrospray ionization mass spectrometry; Fmoc, fluorenylmethoxycarbonyl; HATU, 1-[Bis(dimethylamino)methylene]-1*H*-1,2,3-triazolo[4,5-*b*]pyridinium 3-oxid hexafluorophosphate; LC/MS, liquid chromatography/mass spectrometry; MeOH, methanol; nAChR, nicotinic acetylcholine receptor; NMR, nuclear magnetic resonance; Pbf, pentamethyl-dihydrobenzofuran-5-sulfonyl; RP-HPLC, reversed-phase high performance liquid chromatography; SPPS, solid phase peptide synthesis; t-Bu, tert-butyl;TFA, trifluoroacetic acid; TIS, triisopropylsilane; Tris, 2-Amino-2-(hydroxymethyl)propane-1,3-diol;Trt, trityl; UV, ultra-violet.

### 4.2. Chemical Synthesis

DMF, DIEA, ACN, TIS, TFA, piperidine and all other reagents were obtained from Sigma-Aldrich (Saint-Louis, MI, USA) or Merck (Darmstadt, Allemagne) and were used as supplied. Fmoc (L) amino acid derivatives and HATU were purchased from Iris Biotech (Marktredwitz, Germany). AmphiSpheres™ 20 HMP resin (0.6 mmol/g) and Amphispheres™ 40 RAM (0.4 mmol/g) were purchased from Agilent Technologies (Les Ulis, France). The following side-chain protecting groups were used: Asn(Trt), Cys(Trt), Ser(tBu), Asp(OtBu), Lys(Boc), Arg(Pbf). Peptides were manually synthesized by using the Fmoc-based solid-phase peptide synthesis technique on a VWR (Radnor, PA, USA) microplate shaker. All Fmoc amino acids and HATU were dissolved in DMF to reach 0.5 M. For acid peptides the first residue was anchored on 20 HMP resin using the method described by Grandas et al. [26]. The resin was washed with DMF, DCM, MeOH, and DMF. Fmoc deprotection was carried out with piperidine in DMF (1/2 v/v) twice for 3 min. Subsequent amino acids were coupled onto 0.1 mmol of resin twice for 10 min using an amino acid/HATU /DIPEA ratio of 5:5:10 relative to resin loading. DMF was used for resin washing between deprotection and coupling steps. After chain assembly was complete, the terminal Fmoc group was removed and the resin washed with DMF and DCM. Side-chain deprotection and cleavage from the resin was carried out by adding 10 mL of TFA/TIS/H_2_O (95/2.5/2.5 v/v/v) and stirring the mixture for 2.5 h at room temperature. After the resin was removed by filtration and washed three times with dichloromethane. Dichloromethane and TFA were removed under vacuum then cold diethyl ether was added to precipitate the peptide. The disulfide bridge was formed between the free cysteine residues by dissolving the peptide at 0.2 mM in 50 mM Tris-HCl buffer adjusted to pH 8 and adding dropwise 7 equivalents of DTP at 10 mM in MeOH. When reaction was complete, the reaction mixture was acidified to pH 3 and loaded onto preparative RP-HPLC and pure fractions were combined. The combined pure fractions were freeze-dried and their purity were confirmed by LC/ESI-MS.

### 4.3. Mass Spectrometry

Solvents used for LC/MS were of HPLC grade.

Intermediate products were characterized using a LC/MS system consisting of a Waters (Milford, OH, USA) Alliance 2695 HPLC, coupled to a Waters Micromass ZQ spectrometer (electrospray ionization mode, ESI+). All the analyses were carried out using a Chromolith (Fontenay sous Bois, France) HighResolution RP-18e (4.6 x 25 mm, 15 nm–1.15 µm particle size, flow rate 3.0 mL/min) column. A flow rate of 3 mL/min and a gradient of 0%–100% B over 2.5 min for routine analyses and 0%–30% B over 30 min for quality control of pure products were used. Eluent A: water/0.1% HCO_2_H; eluent B: acetonitrile/0.1% HCO_2_H. UV detection was performed at 214 nm. Electrospray mass spectra were acquired at a solvent flow rate of 200 µL/min. Nitrogen was used for both the nebulizing and drying gas. The data were obtained in a scan mode ranging from 100 to 1000 m/z or 250 to 1500 m/z to in 0.7 s intervals.

Folded peptides were characterized using a Synapt G2-S high-definition MS system (Waters, Corp., Milford, MA, United States) equipped with an ESI source. Chromatographic separation was carried out at a flow rate of 0.4 ml/min on a Acquity H-Class ultrahigh performance liquid chromatography (UPLC) system (Waters, Corp., Milford, MA, United States), equipped with a Kinetex C18 100Å column (100 mm × 2.1 mm, 2.6 mm particle size) from Phenomenex (LE PECQ France). The mobile phase consisted of water (solvent A) and ACN (solvent B) with both phases acidified by 0.1% (v/v) formic acid. Mass spectra were acquired over the range 50 Da to 1800 Da m/z every 0.1 second in the positive ionization mode.

### 4.4. Preparative RP-HPLC

Preparative RP-HPLC was run on a Gilson PLC 2250 Purification system (Villiers le Bel, France) instrument using a preparative column (Waters DeltaPak C18 Radial-Pak Cartridge, 100 Å, 40 × 100 mm, 15 μm particle size, flow rate 50.0 mL/min). Buffer A was 0.1% TFA in water, and buffer B was 0.1% TFA in acetonitrile.

### 4.5. NMR Spectroscopy

Lyophilized synthetic peptides (~1-1.5 mg) were resuspended in 90% H_2_O:10% D_2_O. 2D ^1^H-^1^H TOCSY, ^1^H-^1^H NOESY, ^1^H-^1^H DQF-COSY, ^1^H-^15^N HSQC, and ^1^H-^13^C HSQC spectra were acquired at 290 K using a 600 MHz AVANCE III NMR spectrometer (Bruker, Karlsruhe, Germany) equipped with a cryogenically cooled probe. D_2_O (99.9%) was obtained from Cambridge Isotope Laboratories, Woburn, MA for ^1^H NMR measurements. All spectra were recorded with an interscan delay of 1 s. NOESY spectra were acquired with mixing times of 200-250 ms, and TOCSY spectra were acquired with isotropic mixing periods of 80 ms. Spectra were referenced to the water signal. Two-dimensional spectra were collected over 4096 data points in the f2 dimension and 512 increments in the f1 dimension over a spectral width of 12 ppm. Standard Bruker pulse sequences were used with an excitation sculpting scheme for solvent suppression. Slow exchange experiments were performed by dissolving lyophilized peptide in D_2_O and recording sequential rounds of 1D and ^1^H-^1^H TOCSY spectra and monitoring the amide proton exchange. Spectra were processed using TOPSPIN (Bruker, Karlsruhe, Germany) and analyzed using CCPNMR. The assignments were made using established protocols [27] and the secondary shifts derived by subtracting the random coil αH shift [28] from the experimental αH shifts. The two-dimensional NOESY spectra of the conopressin peptides were automatically assigned and an ensemble of structures calculated using the program CYANA [29]. Torsion-angle restraints from DANGLE [30] were used in the structure calculations. The final structures were visualized using Pymol (The PyMOL Molecular Graphics System, Version 2.0 Schrödinger, LLC.) and MOLMOL [31].

### 4.6. Cell Culture Method and Transient Expression of Human and Zebrafish Oxytocin and Arginine Vasopressin Receptors

The human oxytocin receptor (hOTR), the human arginine vasopressin receptor V1a (hV1aR), the human arginine vasopressin receptor V1b (hV1bR) and the human arginine vasopressin receptor V2 (hV2R) complementary DNAs (cDNAs) were obtained from OriGene Technologies. The corresponding arginine vasopressin zebrafish (ZF) receptors; arginine vasopressin receptor 1Ab (V1a1, The National Center for Biotechnology Information (NCBI) accession number NP_001284605.1), arginine vasopressin receptor 1Aa (V1a2, NCBI accession number NP_001288043.1), arginine vasopressin receptor 2 (V2, NCBI accession number NP_001103595.1) and the oxytocin receptor (OTR, NCBI accession number NP_001186299.1), were synthetically synthesized by GenScript.

COS-1 cells (American Type Culture Collection (ATCC)) grown in Dulbecco’s modified Eagle’s medium (DMEM) and 5% fetal bovine serum (FBS) were transiently transfected with plasmid DNA encoding the hOTR, hV1aR, hV1bR, hV2, ZF V1a1, ZF V1a2, ZF V2 and ZF OTR using FuGENE HD in a 1:3 ratio of DNA and FuGENE, following the manufacturer’s protocol.

### 4.7. FLIPR Assay Measuring Intracellular Ca^2+^ Responses

At 24 h post-transfection, transiently transfected COS-1 cells were seeded at a density of 15,000 cells/well in 384-well black-walled imaging plates (Corning, Sigma-Aldrich) and maintained for another 24 h at 37°C in a 5% humidified CO_2_ incubator. The assay measuring the ligand-induced Ca^2+^ responses was performed 48 h post-transfection. On the day of the assay, cells were loaded with the Calcium 4 No-wash dye (Molecular Devices) by diluting the lyophilized dye in a physiological salt solution (PSS: 140 mM NaCl, 11.5 mM glucose, 5.9 mM KCl, 1.4 mM MgCl_2_, 1.2 mM NaH_2_PO_4_, 5 mM NaHCO_3_, 1.8 mM CaCl_2_, 10 mM HEPES, pH 7.4), and incubated for 30 min at 37°C in a 5% humidified CO_2_ incubator. Intracellular Ca^2+^ responses were measured in response to ligands in a Fluorometric Imaging Plate Reader (FLIPR) (Molecular Devices) using a cooled CCD camera with excitation at 470–495 nm and emission at 515–575 nm. Camera gain and intensity were adjusted for each plate to yield a minimum of 1000 arbitrary fluorescence units (AFU) baseline fluorescence. Prior to addition of control agonists or conopressins, 10 baseline fluorescence readings were taken, followed by fluorescent readings every second for 300 s. Concentration-response curves were established by plotting Delta F/F_0_ values, where F_0_ is the base-line level of fluorescence and Delta F is the change in fluorescence from the baseline level, against agonist concentration using Prism (GraphPad Software). The conopressins were also tested for antagonist activity at the human and ZF receptors. After the addition of 10 µM conopressin peptide, cells were incubated for 10 minutes before stimulating the receptors with an EC_90_ concentration of agonists (oxytocin for the hOTR (0.5 µM), vasopressin for hV1a and hV1b (1 µM), vasotocin for ZF V1a1, V1a2, V2 and OTR (0.1 µM)). Changes in fluorescence responses were assessed for 10 s to set the baseline, then 600 s after addition of antagonist and for a further 300 s after addition of agonists, using the FLIPR as previously described.

### 4.8. LANCE Ultra cAMP Assays

Assays measuring cAMP accumulation were performed 48 hours after transfection following the manufacturer’s instructions (LANCE Ultra cAMP kit, PerkinElmer, Melbourne, Victoria, Australia). To test for agonist activity at the hV2R, increasing concentrations of control agonists or conopressins (10 pM to 100 μM) were added to 500 transfected cells in stimulation buffer in a white 384-well plate (OptiPlate, PerkinElmer Life Sciences). When testing the conopressins for antagonist activity, conopressins (100 μM) were added in the presence of an EC_90_ concentration of vasopressin (0.1 nM agonist) to the cells as previously described. The plates were incubated for 30 min at room temperature. Cells were then lysed by the addition of the europium (Eu) chelate-labeled cAMP tracer and the cAMP-specific monoclonal antibodies labeled with the ULight dye, diluted in cAMP detection buffer (LANCE Ultra cAMP kit, PerkinElmer), followed by incubation for 1 hour at room temperature. The emission signals were measured at 615 and 665 nm after excitation at 340 nm using a Tecan microplate reader (Tecan, Melbourne, Victoria, Australia).

### 4.9. Analysis of the mRNA Pool isolated from Conus miliaris Venom Gland

RNA extraction, cDNA Library, 454 sequencing and assembly has been performed has described by Dutertre et al. [32]. A comprehensive analysis of *C. miliaris* venom transcriptome will be published elsewhere.

## Figures and Tables

**Figure 1 marinedrugs-18-00150-f001:**
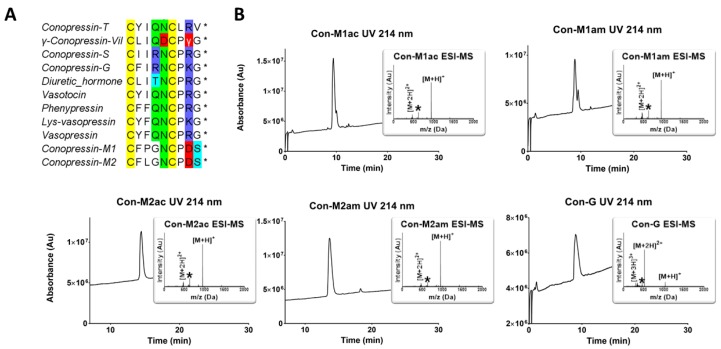
RP-HPLC/ESI-MS analyses of the synthesized conopressins and alignment of conopressin-related sequences. (**A**) Alignment of conopressin-related sequences. The asterisks * indicate an amidated C-terminal. Conopressin-M1 and M2 with γ-conopressin-vil are the only sequences that display a negatively charged amino acid at position 8. Interestingly, conopressin-M1 also displays an unusual proline residue at position 3. The highly conserved glycine residue at position 9 is replaced by a serine residue in conopressin-M1 and M2. (**B**) RP-HPLC/ESI-MS analyses of the synthesized conopressins. Acetonitrile (ACN) gradient from 0% to 30% over 30 min. For Con-M1 the two peaks display the same mass, possibly caused by the two proline residues inducing cis-trans isomerization causing dynamic conformational exchange leading to the splitting of the UV chromatogram peak [16,17,18]. The asterisk (*) on ESI-MS insets indicate an ion resulting from in source fragmentation of the proline residue [19].

**Figure 2 marinedrugs-18-00150-f002:**
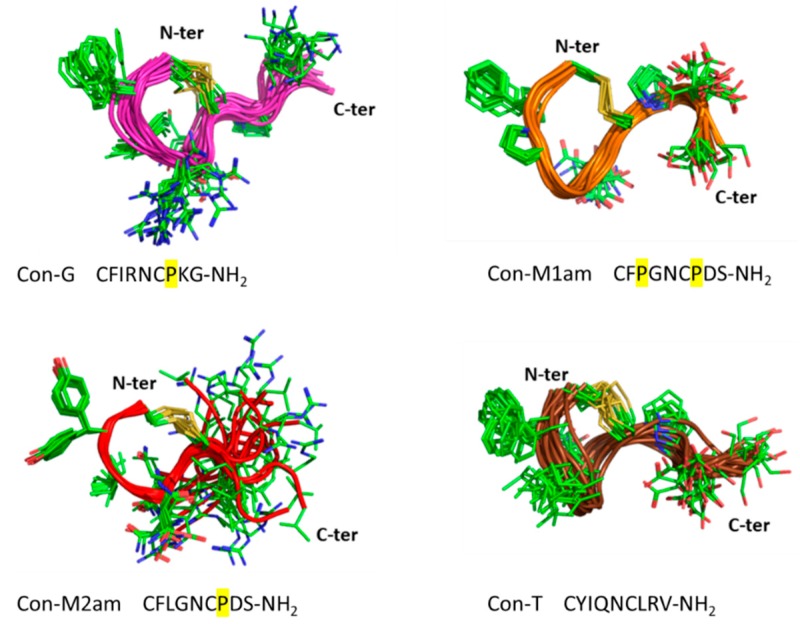
Three-dimensional structures of Con-G, Con-M1, Con-M2 and Con-T. The 20 lowest NMR structures are superimposed over the backbone atoms. The backbone is shown in ribbon format and the side-chains as sticks. Proline residues bringing constraints to the structures are highlighted in yellow.

**Figure 3 marinedrugs-18-00150-f003:**
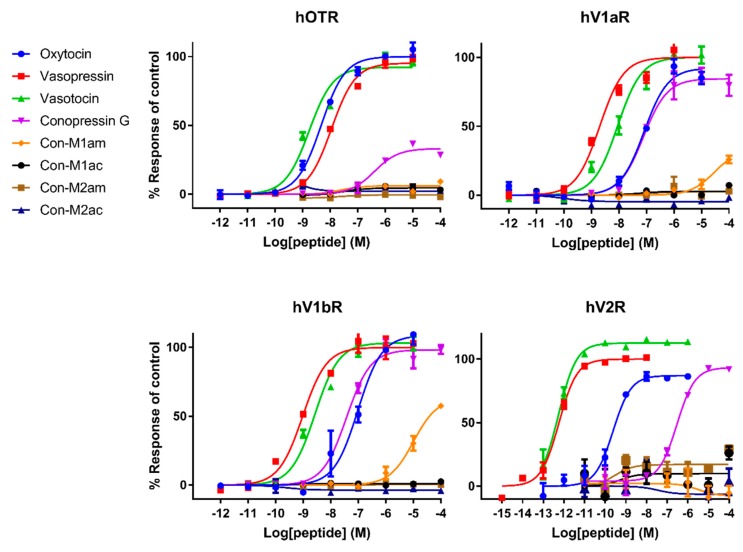
Representative concentration-response curves measuring increasing concentrations of intracellular calcium using a FLIPR assay for the hOTR, hV1aR and hV1bR, and representative concentration-response curves measuring accumulation of cAMP using a cAMP signaling assay for the hV2R of all tested compounds. Each point represents the mean of measurements from one experiment performed in triplicate. Error bars represent S.E.M.

**Figure 4 marinedrugs-18-00150-f004:**
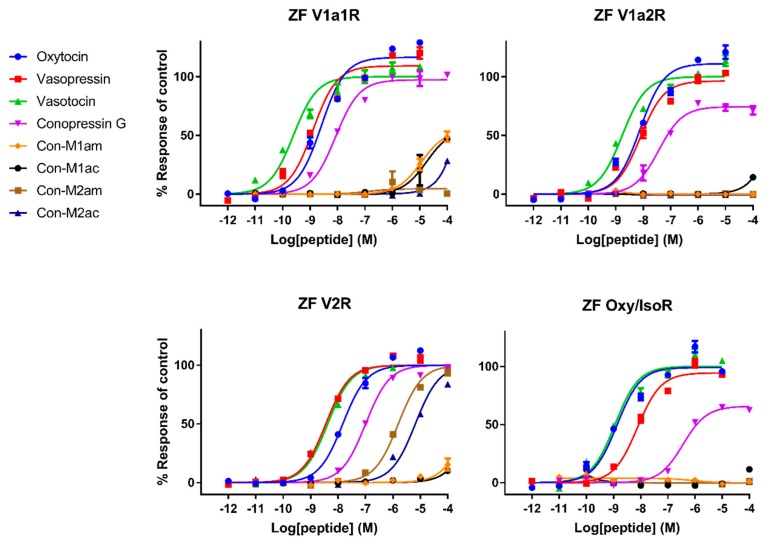
Representative concentration-response curves measuring increasing concentrations of intracellular calcium using a FLIPR assay of all tested compounds against *Danio rerio* (zebrafish) oxytocin-vasopressin related receptors. Each point represents the mean of measurements from one experiment performed in triplicate. Error bars represent S.E.M.

**Figure 5 marinedrugs-18-00150-f005:**
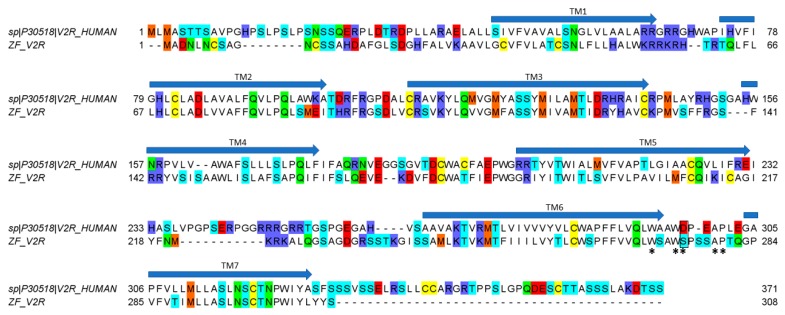
Alignment between hV2R (uniprot entry P30518) and cloned ZF V2R. The substitution of D297 in hV2R with S275 in ZF V2R is bordered in black. Asterisks (*) indicate amino-acid residues that have been suggested to participate and to be important in receptor–ligand interaction. Arrows indicate the seven putative transmembrane domains (TM 1–7).

**Table 1 marinedrugs-18-00150-t001:** Mean EC_50_ (nM) values of all tested peptides on all receptors (min. three independent experiments). Amino acid substitutions relative to vasotocin inducing pharmacological properties differences are highlighted. Arrows indicate lower (down) or higher (up) EC_50_ for the tested peptides relative to vasotocin mean EC_50_ values. Selectivity values are calculated according to the highest EC_50_ value. Boxes indicate significant changes relative to vasotocin. Standard errors of the mean are indicated in brackets. N.D means not determined because of high value > 100 µM.

Name	Sequence	hOTR	hV1aR	hV1bR	hV2R	ZF V1a1R	ZF V1a2R	ZF V2R	ZF oxy/isoR	Selectivity
**Vasotocin**	CYIQNCPRG*	1.62(±0.51)	15.92(±4.57)	4.26(±1.11)	0.00055(±0.0003)	0.41(±0.11)	2.77(±3.33)	5.07(±1.19)	0.85(±0.14)	hV2R (x28945.5) > ZF V1a1R (x38.8) > ZF oxy (x18.7) > hOTR(x9.8) > ZF V1a2R (x5.7) > hV1bR (x3.7) > ZF V2R (x3.1) > hV1aR
**Oxytocin**	CYIQNCPLG*	4.57↗(±1.60)	89.26↗;(±5.12)	84.92↗(±6.44)	0.24↗(±0.14)	3.68↗(±1.70)	8.02↗(±1.44)	15.98↗(±1.64)	1.74↗(±0.75)	hV2R (x371.9) > ZF oxy (x51.3) > ZF V1a1R (x24.3) > hOTR(x19.5) > ZF V1a2R (x11.1) > ZF V2R (x5.6) > hV1bR (x1.1) > hV1aR
**Vasopressin**	CYFQNCPRG*	8.86↗(±2.85)	3.33↘(±0.81)	1.30↘(±0.27)	0.00058=(±0.0004)	1.28↗(±0.19)	8.24↗(±0.28)	4.97=(±0.80)	7.42↗(±0.08)	hV2R (x15275.9) > ZF V1a1R = hV1bR (6.9) > hV1aR (x2.7) > ZF V2R (x1.8) > ZF oxy (x1.2) > ZF V1a2R (x1.1) > hOTR
**Con-G**	CFIRNCPKG*	455.66↗(±39.27)	123.78↗(±27.35)	51.92↗(±8.37)	299.2↗(±11.32)	10.61↗(±1.87)	44.06↗(±7.42)	61.05↗(±12.80)	353.73↗(±12.43)	ZF V1a1R (x42.9) > ZF V1a2R (x10.3) > hV1bR (x8.8) > ZF V2R (x7.5) > hV1aR (x3.7) > hV2R (x1.5) > ZF oxy (x1.3) > hOTR
**Con-M1ac**	CFPGNCPDS	N.D	N.D	N.D	N.D	116 950	N.D	N.D	N.D	
**Con-M1am**	CFPGNCPDS*	N.D	N.D	38 194	N.D	13 614(±9807)	N.D	N.D	N.D	
**Con-M2ac**	CFLGNCPDS	N.D	N.D	N.D	N.D	N.D	N.D	3656(±2173)	N.D	
**Con-M2am**	CFLGNCPDS*	N.D	N.D	N.D	N.D	N.D	N.D	1722(±637)	N.D

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
