# Peer review of "Synthesis, Pharmacological and Structural Characterization of Novel Conopressins from Conus miliaris"

_marinedrugs, 2020, doi:10.3390/md18030150_

Round 1
Reviewer 1 Report
In this paper, Giribaldi et al. identified two novel conopressins from transcriptome of a cone snail, chemically synthesized the peptides, and examined their binding affinity to various human and zebrafish receptors. Their results are novel and interesting with robust data and appropriate discussion. I found no major flaws in this manuscript.
Also, the manuscript is basically well-written in good English, but contains some typos and errors. I pointed some in my minor comments below, but this is not comprehensive. The authors are encouraged to have English proofreading before submitting the revised version of this manuscript.
Major comments
1. Transcriptome screening
The authors described that these conopressins were found from the venom gland transcriptome. I would suggest to include more information about the transcriptome. According to 4.9 it seems that the authors newly performed 454 sequencing for this study. Do they have GenBank accession numbers? How are conopressin sequences identified? - for example using the sequence similarity with known conopressins? Were there any other peptides, or these seem to be only conopressins in this species?
2. Folding condition
Peptides analyzed in Fig. 1B were already folded. Does it mean that the procedure written in L335-337 is the folding procedure? Please clarify. If this is the case, the biochemical condition of the venom gland needs to be mentioned to show that the conopressins are folded under physiological conditions.
3. While the novel conopressins showed a relatively higher affinities to ZF receptors compared to human receptors, the affinity is still kind of low. Is it possible that other GPCR is an endogenous ligand in fish?Minor comments
L93: Change "sequences" to "peptides"? Synthesis of sequences sounds odd.
L111: Capitalize C of conopressin-M1.
L131: Add "and" after however.
L185: "zebrafish (Danio rerio)" would be better to make the expression consistent with other sentences. L388 contains "zebra fish." Please check the consistency throughout the manuscript.
L189: Add (iii) before C. miliaris.
L200: "(Danio rerio)" is not necessary here in my opinion.
L206: Up or down arrows?
L212: C. imperialis: Please make sure that all scientific names in Discussion are in italic.
L315: All other reagents
L331: Add a space between by and filtration.
Table 1, L178, and 416 etc.
Are these human vasopressin? Please provide organisms for control peptides.
Table 1.
Please write the meaning of boxes in "Selectivity"..
Author Response
Reviewer 1
In this paper, Giribaldi et al. identified two novel conopressins from transcriptome of a cone snail, chemically synthesized the peptides, and examined their binding affinity to various human and zebrafish receptors. Their results are novel and interesting with robust data and appropriate discussion. I found no major flaws in this manuscript.
Also, the manuscript is basically well-written in good English, but contains some typos and errors. I pointed some in my minor comments below, but this is not comprehensive. The authors are encouraged to have English proofreading before submitting the revised version of this manuscript.
Major comments
1. Transcriptome screening
The authors described that these conopressins were found from the venom gland transcriptome. I would suggest to include more information about the transcriptome. According to 4.9 it seems that the authors newly performed 454 sequencing for this study. Do they have GenBank accession numbers? How are conopressin sequences identified? - for example using the sequence similarity with known conopressins? Were there any other peptides, or these seem to be only conopressins in this species?
A comprehensive analysis of C. miliaris venom gland transcriptome will be published elsewhere as part of a larger comparative study with other species (now mentioned line 441), but we can reveal that, similarly to other published Conus transcriptomes, C. miliaris produces hundreds of sequences belonging to many different gene families. The focus of this particular study was the unusual conopressins, which were initially retrieved using Conosorter as already described (Lavergne et al., BMC Genomics, 2013). The mature peptide sequences are presented in Figure 1 but the full precursors sequences will be deposited to dedicated database (including GenBank and Conoserver) together with all sequences from this species in the frame of the forthcoming comparative study.
- Folding condition
Peptides analyzed in Fig. 1B were already folded. Does it mean that the procedure written in L335-337 is the folding procedure? Please clarify. If this is the case, the biochemical condition of the venom gland needs to be mentioned to show that the conopressins are folded under physiological conditions.
The procedure described for the chemical synthesis in section 4.2 includes 1) the synthesis of the linear peptide (lines 328-338) and 2) the formation of the unique disulfide bridge (lines 339-342). Given that only two cysteine residues are present in the linear peptides, the formation of the unique disulfide bridge is straightforward, with only one possible conformation. The biochemical condition of the venom gland is poorly known but likely involves chaperone proteins (many disulfide isomerases have been sequenced and characterized from Conus venoms). The most important point here is that our peptides are correctly folded, as shown by MS and NMR.
- While the novel conopressins showed a relatively higher affinities to ZF receptors compared to human receptors, the affinity is still kind of low. Is it possible that other GPCR is an endogenous ligand in fish?
As discussed lines 296-303, we rather suggest that these conopressins have evolved to target worm (prey of C. miliaris) endogenous receptors, likely with higher affinity but this remains to be investigated.
Minor comments
L93: Change "sequences" to "peptides"? Synthesis of sequences sounds odd.
We have replaced “sequences” with “peptides”, as suggested (line 93).
L111: Capitalize C of conopressin-M1.
Thank you for spotting this, we have modified accordingly (line 111).
L131: Add "and" after however.
Added as suggested (line 131).
L185: "zebrafish (Danio rerio)" would be better to make the expression consistent with other sentences. L388 contains "zebra fish." Please check the consistency throughout the manuscript.
We have thoroughly checked the manuscript and systematically replaced “zebra fish” with “zebrafish” (two occurrences, lines 389 and 394).
L189: Add (iii) before C. miliaris.
Thank you, we fixed it accordingly (line 193).
L200: "(Danio rerio)" is not necessary here in my opinion.
We have removed Danio rerio (line 204).
L206: Up or down arrows?
For clarity, we have replaced with “Arrows indicate lower (down) or higher (up) EC50 for the tested peptides relatives to …” line 212.
L212: C. imperialis: Please make sure that all scientific names in Discussion are in italic.
Thank you, we have italicized species names throughout.
L315: All other reagents
Thank you, we have replaced with “all other reagents” line 322.
L331: Add a space between by and filtration.
Modified accordingly line 338
Table 1, L178, and 416 etc.
Are these human vasopressin? Please provide organisms for control peptides.
We are slightly confused as to whether the reviewer refers to the receptors or the peptides for both comments. In any case, it is explicitly stated in section 2.3 (and elsewhere) that the receptors tested are from human and zebrafish. As for the control peptides, these are synthetic in origin, not isolated from an organism. Given that these endogenous peptides were originally isolated from different organisms (i.e. vasopressin is the endogenous peptide found in most mammals, and conopressin-G is found in different cone snail species as well as non-venomous molluscs), we cannot provide such a specific information as “human vasopressin”. Most important, the sequences of all peptides tested are provided in figure 1.
Table 1.
Please write the meaning of boxes in "Selectivity".
Thank you for pointing this out. We have now clarified “boxes indicate significant changes relative to vasotocin”.
Reviewer 2 Report
Dear authors,
In the manuscript by Giribaldi et al., the authors report the discovery of two novel conopressins from the venom gland of Conus miliaris. Pharmacological characterization on human and zebra fish vasopressin and oxytocin indicated low or no agonistic activity on these receptors. However, the unusual amino acid sequences of these novel conopressins as compared to classical vasopressin and oxytocin compounds explained the lack of activity on the respective receptors. The results of these studies should aid in design and synthesis of more selective compounds for human vasopressin/oxytocin receptors.
The manuscript is written very well and the experiments conducted accordingly. I only have some minor suggestions:
- In Table 1, include 95% Confidence Intervals with the EC50 value of each compound
- Same Table, Con-G selectivity ranking seems to be wrong. The rank order should be ZFV1a1R>ZFV1a2R>hV1bR>ZFV2R
- Same Table, if the table will be split in the final version of the manuscript as is in the proof, please include headings for each column on the split table as well.
- On page 8, line 225, fix the type to "profile"
- On page 9, lines 254 and 263, I am assuming the authors are referring to the amino acid residues in the receptors. Not sure why there are decimal points in the amino acid superscript numbers. Please remove those.
- On page 12, line 408, the wavelength values should be "nm" and not "nM".
Author Response
Reviewer 2
Dear authors,
In the manuscript by Giribaldi et al., the authors report the discovery of two novel conopressins from the venom gland of Conus miliaris. Pharmacological characterization on human and zebra fish vasopressin and oxytocin indicated low or no agonistic activity on these receptors. However, the unusual amino acid sequences of these novel conopressins as compared to classical vasopressin and oxytocin compounds explained the lack of activity on the respective receptors. The results of these studies should aid in design and synthesis of more selective compounds for human vasopressin/oxytocin receptors.
The manuscript is written very well and the experiments conducted accordingly. I only have some minor suggestions:
- In Table 1, include 95% Confidence Intervals with the EC50 value of each compound
Whereas we agree that 95% confidence intervals can provide some valuable information to the reader, we have decided to provide instead the SEM, as requested by reviewer 1. Adding both the 95% confidence intervals and SEM would render the already dense table illegible.
- Same Table, Con-G selectivity ranking seems to be wrong. The rank order should be ZFV1a1R>ZFV1a2R>hV1bR>ZFV2R
Thank you for pointing this out. We have now corrected Table 1.
- Same Table, if the table will be split in the final version of the manuscript as is in the proof, please include headings for each column on the split table as well.
On the revised version, the Table 1 is not split and appears on a full page.
- On page 8, line 225, fix the type to "profile"
Thank you, the typo has been corrected line 231.
- On page 9, lines 254 and 263, I am assuming the authors are referring to the amino acid residues in the receptors. Not sure why there are decimal points in the amino acid superscript numbers. Please remove those.
The numbering of receptor residues follows the nomenclature used in the cited publication of Rodrigo et al, where the first number refers to the transmembrane helix of the receptor (1 to 7), and the second number indicates the position of the residue in the helix.
- On page 12, line 408, the wavelength values should be "nm" and not "nM".
We have corrected to “nm” line 414.
Reviewer 3 Report
In Figure 3 and Figure 4, it would be helpful to know without looking at the text in the introduction what is being measured. Some were measuring Calcium and some were measuring cAMP.
Vasopressin receptor is coupled to adenylyl cyclase and V1aR, V1bR and OTR to phospholipase C.
I am not that familiar with this cell system and whether this cells have any endogenous receptors that might be activated. I am assuming there are not. Needs to be mentioned in the results section.
For this reason I think it would have been nice to see a reference that would activate only one of these types of receptors and not the others.
In the discussion when talking about EC50, it would be better to show actual numbers and reference the reader to the most meaningful comparisons. So that I can understand what is being said without having to go over the whole Table #1.
Author Response
Reviewer 3
In Figure 3 and Figure 4, it would be helpful to know without looking at the text in the introduction what is being measured. Some were measuring Calcium and some were measuring cAMP.
Vasopressin receptor is coupled to adenylyl cyclase and V1aR, V1bR and OTR to phospholipase C.
We agree and have added this information in the legends of Fig 3 and 4.
I am not that familiar with this cell system and whether this cells have any endogenous receptors that might be activated. I am assuming there are not. Needs to be mentioned in the results section.
We have added the following sentence in the results, lines 206-207: “...and the control compounds did not induce a response in untransfected cells, which confirms that our responses are receptor specific (data not shown).”
For this reason I think it would have been nice to see a reference that would activate only one of these types of receptors and not the others.
Unfortunately, there are no true subtype specific ligands for these receptors (most known ligands show some selectivity but will activate several receptors, not just one).
In the discussion when talking about EC50, it would be better to show actual numbers and reference the reader to the most meaningful comparisons. So that I can understand what is being said without having to go over the whole Table #1.
We agree and have added relevant EC50 values in the discussion lines 242-251.